# Neuronal Cytoglobin in the Auditory Brainstem of Rat and Mouse: Distribution, Cochlear Projection, and Nitric Oxide Production

**DOI:** 10.3390/brainsci13010107

**Published:** 2023-01-05

**Authors:** Stefan Reuss, Daniel Balmaceda, Mirra Elgurt, Randolf Riemann

**Affiliations:** 1Department of Nuclear Medicine, University Medical Center, Johannes Gutenberg-University, 55101 Mainz, Germany; 2Department of Anatomy and Cell Biology, University Medical Center, Johannes Gutenberg-University, 55099 Mainz, Germany; 3Department of Otorhinolaryngology, Elbe-Kliniken, 21682 Stade, Germany

**Keywords:** cytoglobin, mice, hearing, auditory, rat, immunohistochemistry, brainstem, fluoro-gold, neuronal nitric oxide-synthase, cochlea, oxygen

## Abstract

Cytoglobin (Cygb), a hemoprotein of the globin family, is expressed in the supportive tissue cells of the fibroblast lineage and in distinct neuronal cell populations. The expression pattern and regulatory parameters of fibroblasts and related cells were studied in organs such as the kidney and liver in a variety of animal models. In contrast, knowledge about cytoglobin-expressing neurons is sparse. Only a few papers described the distribution in the brain as ubiquitous with a restricted number of neurons in focal regions. Although there is evidence for cytoglobin involvement in neuronal hypoxia tolerance, its presence in the auditory system was not studied despite high metabolism rates and oxygen demands of the cochlea and related brainstem centers. In a continuation of a previous study demonstrating Cygb-neurons in, inter alia, auditory regions of the mouse brain, we concentrated on the superior olivary complex (SOC) in the present study. We sought to investigate the distribution, projection pattern and neurochemistry of Cygb-neurons in the SOC. We conducted immunohistochemistry using a Cygb antibody and found that this brainstem region, functionally competent for bilateral hearing and providing cochlear hair cell innervation, contains a considerable number of Cygb-expressing neurons (averaging 2067 ± 211 making up 10 ±1% percent of total neuron number) in rats, and 514 ± 138 (6 ± 1%) in mice. They were observed in all regions of the SOC. Retrograde neuronal tract tracing with Fluorogold injected into the cochlea demonstrated that 1243 ± 100 (6 ± 1% of total neuron number in rat SOC)) were olivocochlear neurons. Approximately 56% of total Cygb neurons were retrogradely labelled, while the majority of olivocochlear neurons of both lateral and medial systems were Cygb-immunoreactive. We also conducted double immunofluorescence staining for Cygb and neuronal nitric oxide synthase (nNOS), the enzyme responsible for nitric oxide production, and observed that cytoglobin in the SOC frequently co-localized with nNOS. Our findings suggest that cytoglobin plays an important physiologic role in the oxygen homeostasis of the peripheral and central auditory nervous system. Further studies, also including transgenic animal models, are required to shed more light on the function(s) of Cygb in neurons, in particular of the auditory system.

## 1. Introduction

Cytoglobin (Cygb) is one of eight known members of the globin superfamily, which comprises small proteins that reversibly bind O_2_ to their heme group. While hemoglobin and myoglobin are well studied, knowledge about neuroglobin and cytoglobin, two globins expressed in neurons, is relatively sparse. This applies particularly to cytoglobin (Cygb). Three research groups [1,2,3] discovered the protein independently. It was initially referred to as "stellate cell activation-associated protein" [1] or “histoglobin” [3], but was officially designated as “cytoglobin” [2]. Recombinant Cygb exists as a dimer of two identical subunits of approximately 21 kDa each [4], is hexacoordinated [3], has an average length of 170–190 amino acids, and exhibits an O_2_-affinity of approximately 1 Torr, which is similar to myoglobin [2,3]. 

Cytoglobin is expressed in the cytoplasm of fibroblasts and ontogenetically related cells [5,6]. Various studies, some of which utilized transgenic animal models, demonstrated that Cygb may be able to scavenge free radicals, to protect against fibrosis of liver and kidney, to function as a tumor-suppressing factor, to be involved in tissue regeneration and remodeling, and to play a role in cardiovascular physiology, particularly by the regulation of NO metabolism (for reviews see [7,8,9,10]. A connection between cytoglobin and respiratory function, such as with hemo- and myoglobin, seems rather unlikely due to the low intracellular concentration of Cygb [10,11]. Cytoglobin might instead provide oxygen for O_2_-consuming enzymes, such as for the synthesis of collagen in fibroblasts (cf. [8]). 

Cytoglobin is also found in distinct neuronal subpopulations, where it is located in the cell nucleus, in addition to the cytoplasm [6]. It was speculated that Cygb transfers signals into the nucleus to affect gene expression [12]. Although in most studies merely restricted regions in single species were explored, there is agreement that Cygb-neurons are present throughout the central nervous system but expressed only in a restricted number of neurons in focal regions [6,13,14,15,16]. Detailed mapping of Cygb-positive neuronal cells has only been performed in mouse brains so far [16,17]. 

Which, if any, of the above-mentioned Cygb-functions are also active in neurons is open, since the majority of studies did not examine brain structure or metabolism. It was shown, however, that in the brain of the subterranean mole rat Cygb is clearly elevated in both normoxia and hypoxia compared to rats, suggesting involvement in neuronal hypoxia tolerance [11].

Our detailed report on the distribution of cytoglobin in the mouse brain [17] demonstrated that auditory cortical and subcortical sites such as the medial geniculate nucleus, inferior colliculus, cochlear nucleus and superior olivary complex (SOC) display distinct populations of Cygb-immunoreactive neurons. Further analyses or characterizations have not been performed so far. We, therefore, conducted a comprehensive analysis of cytoglobin-neurons in the SOC in rats and mice. The SOC is an important relay center for bilateral hearing in the ascending auditory pathway, and provides cochlear hair cell innervation via efferent fibers (cf. [18,19]).

We now report that Cygb-neurons are found in considerable numbers in all SOC subnuclei in both species, that these include olivocochlear neurons, and that many express neuronal nitric oxide-synthase, the enzyme responsible for NO synthesis.

## 2. Materials and Methods

### 2.1. Animals

The experiments were conducted on twelve adult male Sprague Dawley rats (b.wt. 220–250 g) and eight male Balb/C-mice, aged two months (b.wt. 23–26 g). They were bred and held under constant conditions (12:12 hours light–dark rhythm, food and water ad libitum, room temperature 21 ± 1 °C) in the animal facility unit of the Department of Anatomy and Cell Biology, Johannes Gutenberg University, Mainz, Germany. Experimental protocols (animal housing and neuronal tracing) were approved by the local Administration District Official Committee (Bezirksregierung Rheinhessen-Pfalz, Az 177-07/961-30) and were in accordance with the published European Health Guidelines. All efforts were made to minimize the number of animals and their suffering.

### 2.2. Olivocochlear Neuronal Tracing

For cochlear injections, rats were anesthetized with tribromoethanol (0.3 g/kg b.wt. i.p.). The cavity of the middle ear was exposed by a dorsal approach. The auditory ossicles were disconnected, the round window punctured and a small amount of perilymph was removed. Then, the tracer (approximately 200 nl of a 5% FG solution (Fluorochrome, Englewood, CO, USA; dissolved in distilled water) was injected into the Scala tympani using a glass micropipette of approximately 3 μm tip diameter connected to a Hamilton syringe. The round window was sealed with bone wax. After 5 days, rats were killed by ether overdose and perfusion-fixed as described below.

### 2.3. Tissue Processing and Immunofluorescence Incubations

Animals were killed by ether overdose at the middle of the light period and immediately perfused transcardially with PBS to which 15,000 IU heparin/L was added, at RT, followed by an ice-cold paraformaldehyde-lysine-periodate solution (PLP; [20]). The right atrium was opened to enable venous outflow. 

For brainstem immunohistochemistry, PLP-perfused rat and mouse brains were removed, marked on one side, postfixed in PLP for 1 h, and stored overnight at 4 °C in phosphate-buffered 30% sucrose for cryoprotection. They were then sectioned serially at 40 µm (rats) or 30 µm (mice) thickness on a freezing microtome in the frontal plane and collected in PBS. Non-specific binding sites were blocked at room temperature (RT) for 1 h with 1% bovine serum albumin in PBS. Sections were then incubated free-floating overnight at RT with custom-made rabbit-raised polyclonal antibodies directed against the amino acid positions 2 to 16 of the N-terminus of the Cygb protein (1:1000 in PBS, see [6] for further antibody details). Then, 1% normal donkey serum and 0.1% Triton-X 100 were added to the incubation medium. 

The sections were then washed 3 × 10 min in PBS and incubated for 90 min at RT in the dark with the secondary antibody (Cy3-conjugated F(ab′)2-fragments of goat anti-rabbit IgG, 1:300 in PBS, Dianova, Hamburg, Germany). Sections were mounted on gelatinized glass slides, dried, cleared in xylene, and cover-slipped with Merckoglas (Merck, Darmstadt, Germany).

### 2.4. Double Immunofluorescence Incubation of Mouse Brainstem Sections

For double-immunostaining experiments, sections were incubated in primary Cygb-antibody as described above and, simultaneously, in a polyclonal sheep antibody raised against neuronal nitric oxide-synthase (nNOS, 1:50, Abcam, Cambridge, UK; Cat# ab6175, lot 152546, RRID:AB_305343). The antibody was used and characterized in our laboratory previously [21]. The additional primary antibody was visualized using Cy2-conjugated F(ab′)2-fragments of anti-sheep IgG (Dianova) at 1:300 dilution. Sections were washed and further processed as described above.

### 2.5. Cell Quantifications and Statistical Analysis

For statistical analysis using the Desmos scientific calculator, Cygb-cells were counted from each SOC section from five rats. Neurons exhibiting immunoreactivity to the antibodies tested were counted when labeling was clearly above background level. Numbers were corrected according to [22] to prevent double counting of cells. The factor was separately calculated for each group depending on average cell size and section thickness.

The mean numbers ± standard deviations (SD) were calculated for each SOC region. The percentages of Cygb neurons of the total number of neurons were calculated for each SOC region. From an additional group of five rats with retrograde tracing of olivocochlear neurons, the numbers of FG-cells, of Cygb-cells, and of FG/Cygb-neurons were counted from each SOC region and the means per animal ± SD were calculated. Double-labelled neurons were also calculated as percentages of Cygb-neurons and of FG-neurons (means ± SD per SOC region).

In mice, numbers of Cygb-cells and those of Cygb/nNOS-cells were counted from each SOC section from five animals for each SOC region separately. The percentage of double-labelled cells was calculated for each region of each animal, and the means ± SD of cell numbers and the percentages of double-labelled Cygb-neurons were calculated per region from five animals.

Brainstem regions were delineated according to the stereotaxic brain atlases for rat and mouse [23,24], and the overviews given by [25,26]. 

### 2.6. Image Analysis

Sections from rat and mouse brainstems (neuronal tracing and immunofluorescence material) were analyzed using an Olympus BX51 research microscope equipped with an epifluorescence unit and highly specific single and dual band filter sets, allowing the single or simultaneous axcitation and observation of dyes without overlapping artifacts (Olympus fluorescence monochromatic and dichromatic mirror cubes for FG (excitation center wavelength 360 nm, bandwidth 50 nm), Cy2 (480/40 nm) and Cy3 (540/25 nm). Images were taken and stored with an Olympus ColorView-12 digital color camera operated employing the AnalySIS image processing program (Olympus Soft Imaging Solutions, Münster, Germany). Images were converted into grayscale to achieve higher contrast. The Adobe Photoshop and Powerpoint programs were used to arrange digital images, to adjust image contrast and brightness of whole images, and to add labels.

### 2.7. Control Studies 

Control incubations showed that blocking the antisera with the respective antigen as well as the omission of primary and/or secondary antisera resulted in the absence of respective staining, and that no cross-reactivity between primary and secondary antibodies was present. Negative control experiments which substituted normal rabbit and mouse IgG for the primary antibodies resulted in the complete absence of staining.

Tracing controls consisted of additional rats in which FG was applied to areas outside the Scala tympani. In these cases, the tracer was not observed in the superior olivary region. 

## 3. Results

### 3.1. Localization of Cytoglobin Protein in Rat Brain

The general distribution of cytoglobin in the brain was as described previously for rats, mice, and men [6,11,16,17]. Its presence in auditory structures, however, has been described so far only cursorily in our analysis of the mouse central nervous system [17]. 

Within the auditory system, immunofluorescence of differential intensity was observed in the auditory cerebral cortex, medial geniculate body, inferior colliculus, lateral lemniscal nuclei, cochlear nucleus and superior olivary complex (SOC), as described in detail earlier [17]. While cytoglobin was consistently detected in the cytoplasm of neuronal somata and processes, and in their cell nucleus, the antibody did not label glial cells. Neurons of the SOC were subjected to further analysis in the present study.

In the rat auditory brainstem, Cygb-immunofluorescence was observed in neurons in the superior olivary complex (overview given in Figure 1A). Throughout the SOC, a substantial portion of neurons was Cygb-immunolabelled (Figure 1A,B). A total of 2067 ± 211 (mean ± SD, rounded) Cygb neurons per animal were counted from five rats. Using the averaged total numbers of neurons in the subnuclei of the rat SOC [27] we calculated the percentage of total Cygb-immunoreactive neurons to be 10 ± 1%. The percentages of Cygb-neurons in SOC regions ranged from 7 ± 2 (mean ± SD, rounded) in the MSO to 27 ± 2 in the VPO. A comparison is given in the diagram (Figure 1B), and examples are presented in Figure 1C–G).

### 3.2. Cygb-Immunoreactivity in Identified Olivocochlear Neurons in the Rat

To study the relation of Cygb-neurons and cochlear efferent neurons, we injected Fluorogold (FG) unilaterally into the rat Scala tympani. Following retrograde neuronal transport, the tracer was consistently found in neuronal perikarya and processes of the bilateral SOC. On average, 1243 ± 100 neurons per animal were labelled in the SOC, which made up approximately six percent of the total number of neurons (taken from [27]). 

They were seen in three topographically separated groups, as is typical for OC neurons [28]. Lateral OC (LOC) neurons within the borders of the LSO (Figure 2A) were small and spindle-shaped and were located nearly exclusively ipsilaterally. The second group of LOC neurons, i.e., shell-neurons, was located around the LSO and in periolivary regions (Figure 2B,E). Shell neurons were large and their processes were oriented circularly around the LSO. Ninety percent of shell neurons were located ipsilaterally to the injection site. Medial olivocochlear (MOC) neurons, as the third group, were seen predominantly in rostral and ventral periolivary areas with a light contralateral dominance. MOC neurons were also observed in the ventral aspect of the posterior part of the MNTB. 

Sections containing retrogradely labelled neurons were incubated with the Cygb antibody. We found that approximately 93% of all identified OC neurons were Cygb-positive (Figure 2H), while only small differences between SOC regions were observed. In contrast, the proportion of FG-labelled neurons within the population of Cygb-immunolabelled neurons in these regions was smaller (56% in average, ranging from 32 ± 3 in the LSO to 95 ± 2 in the MNTB; see Figure 2G). In detail, 32% of the Cygb-neurons in the LSO were labelled by FG (and thus were olivocochlear neurons).

### 3.3. Cytoglobin in the Superior Olivary Complex of the Mouse

The distribution of Cygb-immunofluorescent neurons in the mouse brain was as described previously [17], and was also widely identical to the pattern observed in rats (see above). Cytoglobin was expressed in neuronal cells of all areas of the mouse’s superior olivary complex. Examples are given in Figure 3. Small Cygb-immunoreactive cells were observed in the LSO, and were slightly more abundant in the LSO´s lateral limb. The MSO has Cygb-immunoreactive neurons arranged in the dorsoventral axis. Furthermore, immunoreactive neurons were observed in peri- and paraolivary regions, as well as in the MNTB (Figure 3F). The quantification showed a total number of 514 ± 138 Cygb-labeled neurons in the SOC (mean of five animals ± SD).

### 3.4. Colocalization of Cytoglobin and Neuronal Nitric Oxide Synthase in the Mouse SOC

In order to further characterize Cygb-positive neurons in the SOC, we employed double-immunolabeling for Cygb and nNOS, the enzyme responsible for NO synthesis in neurons. Double-labelled neurons were found in all regions of the mouse SOC. Examples are presented in Figure 3. The overview of the LSO (Figure 3A,B) shows that the distribution of both proteins are different, and that approximately twice as much nNOS- than Cygb-neurons was present. There was more overlap between these two neuronal subpopulations in the lateral than in the medial LSO limb. A higher magnification of the boxed region in Figure 3A demonstrates some cytoglobin/nNOS-positive neurons in the lateral limb (exemplarily marked "1"), a considerable number of nNOS-positive/Cygb-negative neurons, and neurons that express only Cygb (“2”). Relatively, many of the large neurons of the MNTB exhibit Cygb immunoreactivity, and the same is true for nNOS immunoreactivity. 

Accordingly, the degree of colocation of both substances in MNTB neurons is relatively high (Figure 3E–H). However, in the marginal regions, some cell groups are Cygb-positive (marked by an arrow in Figure 3E) but nNOS-negative. In the medially located regions of the MNTB, mainly double-immunoreactive neurons are present (marked by "1" in Figure 3G,H), but also cells containing only Cygb ("2") were seen. Neuronal NOS-positive, Cygb-negative cells were not observed in the MNTB. The percentages of double-labeled neurons in relation to Cygb-neurons was 75 ± 16% on average and varied from 50 ± 4% in the LSO to 88 ± 8 in superior and ventral periolivary regions (Figure 3I).

## 4. Discussion

In the present study, we sought to investigate the distribution and some characteristics of neurons expressing cytoglobin in the superior olivary complex (SOC) of rats and mice. Cytoglobin is a respiratory protein expressed in cells of the fibroblast lineage and in distinct neuronal populations. Using fluorescence immunohistochemistry and neuronal tract tracing, we demonstrate that a considerable portion of SOC neurons exhibit Cygb-immunofluorescence, that these include cochlear efferent neurons and that some co-express neuronal nitric oxide-synthase.

In the brain, Cygb-neurons exhibited a differential distribution, with dense clusters in some regions and fewer neurons in others. There is agreement that Cygb-immunoreactive neurons are present throughout the brain in rats, mice, and men, although in most studies only restricted regions were investigated [6,13,14]. A Cygb expression study of the mouse brain [15] using several quantitative methods yielded similar results to the immunostaining studies.

Two features of Cygb-immunoreactive brain cells should be emphasized. First, while cytoglobin was consistently detected in neurons, the antibody did not label glial cells. There is further no evidence from the literature that glial cells may express Cygb, while the related protein, neuroglobin, is also expressed by astrocytes under special physiological conditions [11,29]. Second, in agreement with previous observations, we detected Cygb immunoreaction in the cell soma (cytoplasm and nucleus) and in processes (dendrites and axon) in all neural regions investigated. This is in contrast to non-neuronal cells of the fibroblast lineage, which do not show nuclear staining. A "universal" function of cytoglobin in metabolism appears, therefore, unlikely. 

The location of Cygb in the neuronal nucleus indicates the presence of a neuron-specific factor that allows active transport or passive diffusion of the protein from cytoplasm to nucleus. The nuclear location was confirmed by cell fractioning and Western blot analysis [30], and by living cell imaging of cytoglobin fused to the nuclear localization factor [31]. The role of Cygb particularly in the nucleus lacks functional interpretation, but this phenomenon is thought to be related to an additional specific function of Cygb in neuronal tissue. It was considered that the location of Cygb protein in the cytoplasm and nucleus of the neuron indicates a sensing function. Cytoglobin could act as an O_2_ or NO sensor mediating between the cytoplasm and nucleus to modulate adaptation processes of neurons [12]. A role of Cygb in NO metabolism remains conceivable because of the relatively high degree of colocation with nNOS (see Section 4.4). 

With regard to the location of Cygb-neurons, detailed analyses are available only for mice [16,17]. Both studies agree on a widespread distribution of the protein in a limited number of cells. In our mouse study, we addressed auditory structures as well, and observed Cygb-immunofluorescence of differential intensity in the auditory cerebral cortex, medial geniculate body, inferior colliculus, lateral lemniscal nuclei, cochlear nucleus and superior olivary complex (SOC). In the present study, we conducted further analysis of neurons of the SOC. This group of interrelated nuclei provides olivocochlear regulation and binaural hearing (cf. [18,19]).

### 4.1. Distribution of Cytoglobin in the Superior Olivary Complex

We observed Cygb-immunoreactivity in all regions of the SOC. In average, ten percent of the SOC neurons expressed Cygb. Their portion ranged from 7% in the MSO to about one-third in the ventral periolivary regions, where mainly large neurons are located. There was no indication that their morphological or anatomical appearance is different from Cygb-negative neurons, nor is it known how they physiologically contribute to the hearing process. However, about half of the Cygb-neurons exhibit a cochlear-efferent projection pattern (see Section 4.2). Whether the remaining have intrinsic function, and/or project to other auditory brain sites such as the inferior colliculi or the dorsal nucleus of the lateral lemniscus (cf. [25]), will be the subject of future studies. 

The number and distribution of Cygb-neurons were similar in the SOC of mice (see Section 4.3), while corresponding data from other species are lacking. Interestingly, similar patterns have been observed for neuroglobin but it is not known whether both globins are expressed in the same SOC neurons. In other brain regions, however, such as tegmental nuclei, neurons double-labelled for Cygb and Ngb were found [16].

### 4.2. Cytoglobin Expression by Rat Olivocochlear Neurons

In order to test whether identified olivocochlear neurons are Cygb-immunoreactive, we injected the retrograde neuronal tracer FG into the Scala tympani of rats, as conducted in several previous studies in our laboratory (e.g., [27,32,33]). After uptake by terminals in the organ of Corti and retrograde transport of FG, we found labelled olivocochlear neurons (OCN) in the bilateral SOC. Their number and distribution were consistent with previous reports (e.g., [27,33,34,35,36]).

By combining FG-tracing and Cygb immunohistochemistry, we found that approximately all Cygb neurons in the MNTB exhibited FG and were thus identified as olivocochlear neurons, while the corresponding portions were clearly smaller in LSO, MSO and periolivary regions (see Figure 2G). On the other hand, a large portion of the FG-labelled olivocochlear neurons expressed Cygb, with only little variation between SOC regions. These high levels of coexistence of FG and Cygb (>80%) were observed throughout the SOC, suggesting that many OCN of both the lateral (LOC) and the medial (MOC) olivocochlear system express cytoglobin. 

Cell bodies of the former are located in or around the LSO and their axons provide efferent synapses onto inner hair cell (IHC) afferents, whereby they modulate the glutamatergic IHC-afferent terminal synapses, and thus regulate the activity of type I spiral ganglion neurons in response to sound stimulation (cf. [28,37,38]). Since both LOC neuron groups express Cygb to similar degrees, we will not attach importance to their different projection patterns [39]. MOC neurons are located in rostral and ventral periolivary areas and provide efferent synapses directly onto outer hair cells (OHC), where they convey the cochlear amplifier mechanism, the modulation of otoacoustic emissions, and protection against acoustic injury [28,39,40,41]; for review, see [18]. Since both LOC and MOC neurons express Cygb, it may be assumed that they participate in the important role of the SOC in protecting hair cells from sound-induced damage. 

An interesting question is whether these neurons also transport Cygb mRNA and/or protein into the organ of Corti. The single report on Cygb-mRNA in the cochlea [42] leaves open whether it stems from spiral ganglion neurons, cochlear fibroblasts and pericytes, or OCN axons. There is, however, increasing evidence for mRNA transport and translation in axons to allow rapid responses to physiological needs (cf. [43]). Neuroanatomical evidence for axonal Cygb-transport over relatively long distances was found previously by the intense fiber staining observed, inter alia, in the fasciculus retroflexus (Meynert), and in olfactory and optic nerves [17].

### 4.3. Distribution of Cytoglobin in the Mouse Superior Olivary Complex

In our study of the mouse brainstem, we observed distinct cytoglobin labeling of neuronal subpopulations in almost all areas of the superior olive complex. Visual assessment of fluorescence registers more pronounced signals of the Cygb-antibody in the SOC compared with the surrounding tissue, but this may mirror the higher cellular density in the SOC. Distinct Cygb-immunoreactivity is present in the main nuclei of the superior olive complex (MSO, LSO), particularly in the lateral (low frequency) limb of the LSO. In addition, the medial nucleus of the trapezoid body (MNTB), as well as other periolivary cell groups (VPO, DPO), shows numerous Cygb-immunoreactive cells.

While the portion of Cygb-neurons in the rat SOC were calculated to be ten percent (see above), we could not easily determine the respective percentage in mice since the total numbers of mouse SOC neurons are not available from the literature. The approximate Cygb cell numbers, as determined in the present study, indicate a 4:1 ratio (2000 per rat, 500 per mouse). The SOC volumes of rat and mouse, as determined from the atlases [23,24], exhibit an approximate 5:1 ratio, fitting well to the 5:1 ratio of average brain weights (2 g in rats, 0.4 g in mice; see [44,45]). Since the total neuron number of the rat SOC averages about 20.000 [27,46], approximately four to five thousand SOC neurons in the mouse may be estimated. The number of 500 Cygb neurons of the mouse SOC would thus reveal that, as in rats, one tenth of total SOC neurons expressed Cygb in mice.

### 4.4. Colocalization of Cytoglobin and Neuronal Nitric Oxide-Synthase in the Mouse Superior Olivary Complex

Evidence from previous studies that Cygb-neurons may co-express neuronal nitric oxide-synthase (nNOS) in some brain regions [11,16,47] prompted us to compare the distributions of Cygb and of nNOS, the enzyme responsible for the synthesis of the gaseous neuroactive substance nitric oxide (NO), in the mouse SOC.

The presence of nNOS in SOC and the cochlea was first described nearly three decades ago [48,49], and since then characterized in several studies (for reviews see [50,51]). We observed a considerable portion of cells that exhibited colocation of both substances. The percentages of Cygb neurons that were also nNOS-immunoreactive ranged from approximately 50 in the LSO to >80 in periolivary regions, and averaged 74% of total Cygb cell number. This relatively high portion of neurons expressing Cygb and nNOS indicates that the substances interact in functions such as delivering oxygen during NO production or scavenging excess nitric oxide. Notably, it was demonstrated that Cygb has NO dioxygenase activity, i.e., it can convert the inter- and intracellular signaling molecule NO to nitrate to render NO, which is cell damaging at high concentrations, harmless [52,53]. The colocation of NO and Cygb may argue for the involvement of Cygb in NO metabolism (cf. [8]).

### 4.5. Possible Function of Cytoglobin in Auditory Brainstem 

The physiological role of Cygb in neurons is still not understood to date (cf. [7]). Notably, much information on molecular function comes from animal models in which knockout, silencing, or overexpression were implemented, and this also applies to globins (cf. [54]). However, the effects of Cygb manipulations were investigated in several studies on organs such as kidney or liver (cf. [7]), while the possible effects on brain or neuronal parameters remain enigmatic. To the best of our knowledge, the possible neuronal effects of abolished or attenuated expression of Cygb on morphological or functional parameters have not been described to date. This applies to the mammalian brain in general, and thus to the SOC (and cochlea) as well. In contrast, the characterization of neuroglobin, a related globin, in the cochlea and auditory brainstem [33], was followed by the exploration of knockout effects on hearing and recovery after acoustic trauma [55].

Since cells of fibroblastic origin have a distinct capacity for collagen synthesis, which is an intensively O_2_-consuming process, it was supposed that Cygb supplies O_2_ to prolyl-4-hydroxylase, allowing the hydroxylation of proline residues to procollagen. Collagen production, as well as Cygb expression, similarly increases under hypoxia [6,15]. Furthermore, in active forms of fibroblast-related cells, such as osteoblasts and chondroblasts, the concentration of Cygb is higher than in mature forms, such as osteocytes and chondrocytes, which no longer synthesize collagen [6]. A link between collagen synthesis and Cygb in fibroblasts and related cells is thus likely. Although this process was not usually associated with neurons, there is increasing evidence for the role of “unconventional” (transmembrane) collagens in neural circuit formation and function in both region- and cell-specific manners (cf. [56,57]), rendering an analogous function in neurons possible. 

There are, however, several further possible mechanisms by which cytoglobin may influence auditory functions. First, it may support the metabolism of SOC neurons that, to a considerable amount, synthesize this globin. This may be particularly the case in times of local hypoxia.

Second, under the assumption that Cygb serves as an oxygen-sensing molecule, it may be involved in the regulation of cochlear microcirculation. Following transport into the cochlea by olivocochlear neurons (see Section 4.2), Cygb expression would be upregulated upon local hypoxia mediated by the hypoxia-inducible factor (HIF)-1α. This may result in augmented vascular endothelial growth factor (VEGF) and superoxide dismutase activity, subsequent increase of microvessel diameter [9,10,58], and thus improved blood supply of the organ of Corti.

Third, cochlear Cygb may not or not solely originate in the brainstem and, additionally, be expressed by cochlear pericytes, i.e., contractile cells attached to the outside of capillaries. These exert an important function in the regulation of cochlear microcirculation (cf. [59]). Hypoxic conditions in the cochlea may result in Cygb upregulation and the relaxation of pericytes.

Whatever the source, distribution and function of cytoglobin are, it is among the genes most markedly upregulated in the cochlea of senile rats, and it is the gene most strongly correlated with age-associated changes in auditory brainstem response (ABR) and distortion-product otoacoustic emissions (DPOAE) [42]. Future studies are required to elucidate the functional role of Cygb-expressing neurons of the SOC. Aside from projection patterns and neurochemical environment, immunofluorescence studies are needed to investigate which transmitter-systems may regulate Cygb-neurons, Cygb-OCN and Cygb/nNOS-neurons in the lower auditory brainstem.

## 5. Limitations of the Study

The most clear-cut limitation is that the neuroanatomical results are predominantly descriptive and do not provide functional data. In this early stage of nerve globin research, in particular with regard to cytoglobin, knowledge about its expression in neurons does not include quantitative data. It is also not known how Cygb-expression is regulated in the auditory brainstem. Quantitative studies utilizing in situ-hybridization or polymerase chain reaction are still missing. These may relate to effects of aging and noise-related hearing loss, or recovery after trauma or hypoxia. It would be of particular interest to know how these proceed in animals in which Cygb is overexpressed or its gene silenced. Functional parameters such as ABR and DPOAE may help to overcome these limitations. 

## 6. Conclusions

Our study provides the first analysis of cytoglobin expression in the superior olivary complex of the auditory brainstem in two rodent species. We demonstrate, by single and double immunohistochemistry and retrograde neuronal tracing, that Cygb is highly expressed in rat and mouse SOC neurons, including rat olivocochlear neurons, and that many of these co-express neuronal nitric oxide-synthase. It is expected that, based on the neuroanatomical data presented here, further studies will add functional understanding of the role of this globin in auditory processing. Lesions of the bundle of Rasmussen may indicate whether Cygb is transported into the cochlea by olivocochlear neurons, or whether it is produced by cochlear cells such as pericytes. Cytoglobin-deficiency and overexpressing animal models may further elucidate the role of Cygb in the superior olivary complex and cochlea. This may help to develop new clinical approaches to prevent cochlear oxidative stress, which is thought to be a leading cause of sensorineural hearing loss and other auditory dysfunctions, such as tinnitus [60].

## Figures and Tables

**Figure 1 brainsci-13-00107-f001:**
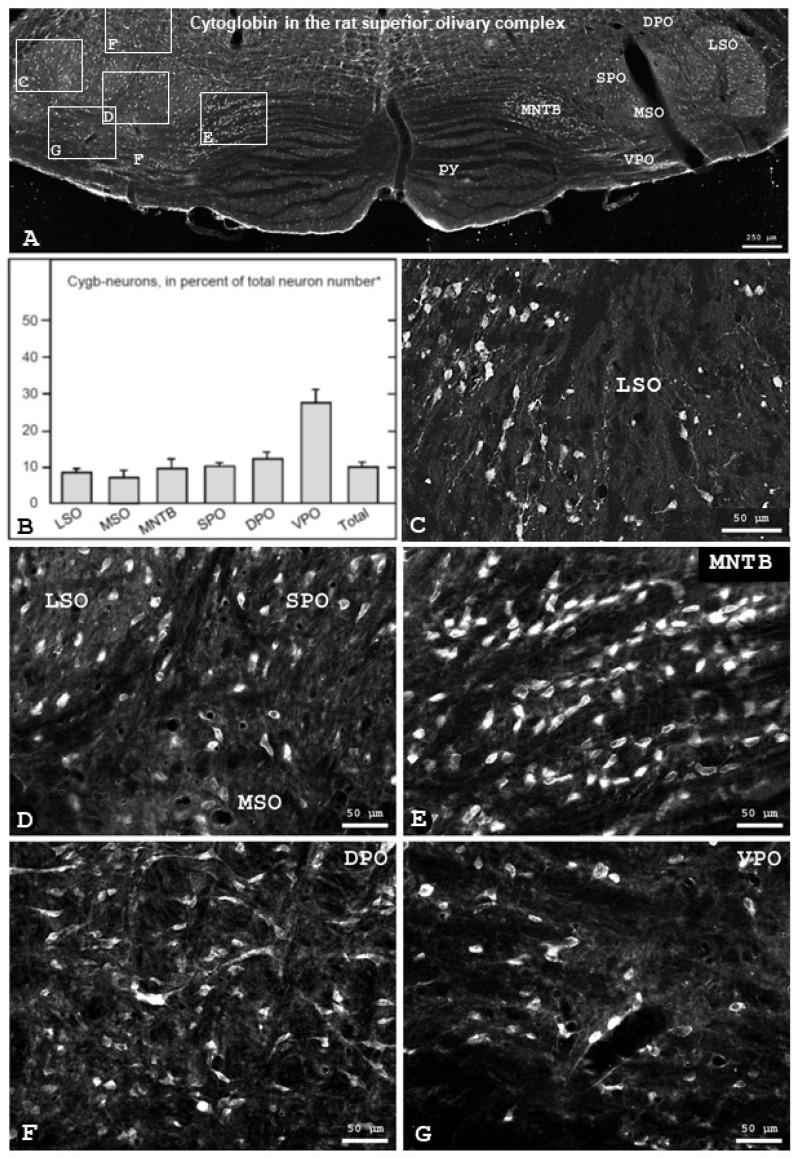
Cytoglobin (Cygb) protein distribution in a frontal section of the rat superior olivary complex (SOC). (**A**) Overview of Cygb-immunofluorescence in the bilateral SOC, taken from an intermediate section of the brainstem, demonstrating labeled neurons in all SOC regions. (**B**) depicts the numbers of Cygb-neurons in percent of the total neuron number for SOC regions (mean ± SD, n = 5), * taken from [25]. (**C**) shows the LSO in a section of the caudal SOC, demonstrating labeled spindle-shaped neurons and their processes oriented in dorsoventral direction. Higher magnifications from the left SOC shown in A demonstrate immunopositive neuronal perikarya in LSO, SPO, MSO (**D**), MNTB (**E**), DPO (**F**) and VPO (**G**)**.** Orientation in each section is medial left, dorsal up. Abbreviations: see list.

**Figure 2 brainsci-13-00107-f002:**
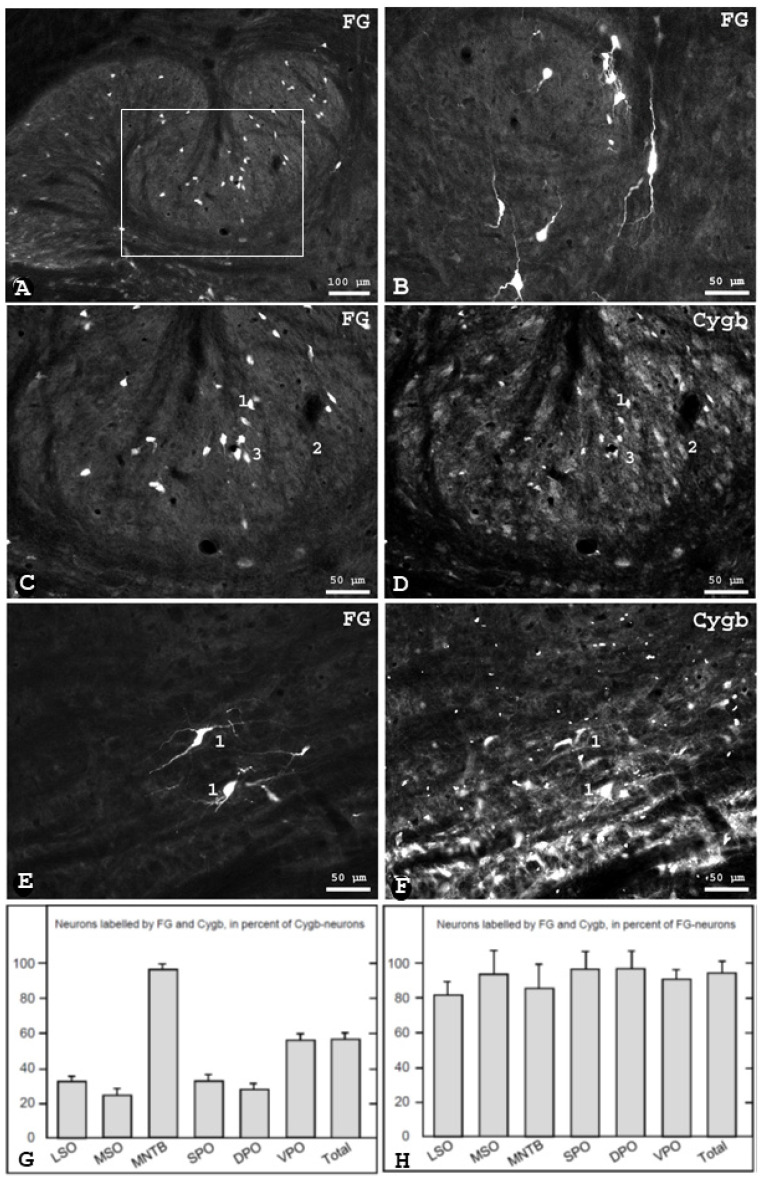
Labelled neurons in the ipsilateral superior olivary complex of the rat after injection of Fluorogold (FG, shown in **A**–**C**) into the cochlea and retrograde axonal transport, combined with cytoglobin (Cygb) immunofluorescence, shown in (**D**,**F**). The boxed area in A is shown in (**C**,**D**), where many double-labelled cell bodies (exemplarily marked “1”), but also neurons solely labelled by Cygb (“2”) or FG (“3”), were observed. Large shell neurons identified by FG (“1”) are shown in (**B**), and in (**E**) belonging to a group of large Cygb-neurons in the VPO in (**F**). The Cygb distribution pattern was not affected by cochlear injection. (**A**–**F**) frontal plane, orientation: medial left, dorsal up. Abbreviations: see list. (**G**): olivocochlear neurons identified by FG in percent of Cygb neurons. (**H**): Cygb-neurons in percent of olivocochlear neurons (mean ± SD, *n* = 5).

**Figure 3 brainsci-13-00107-f003:**
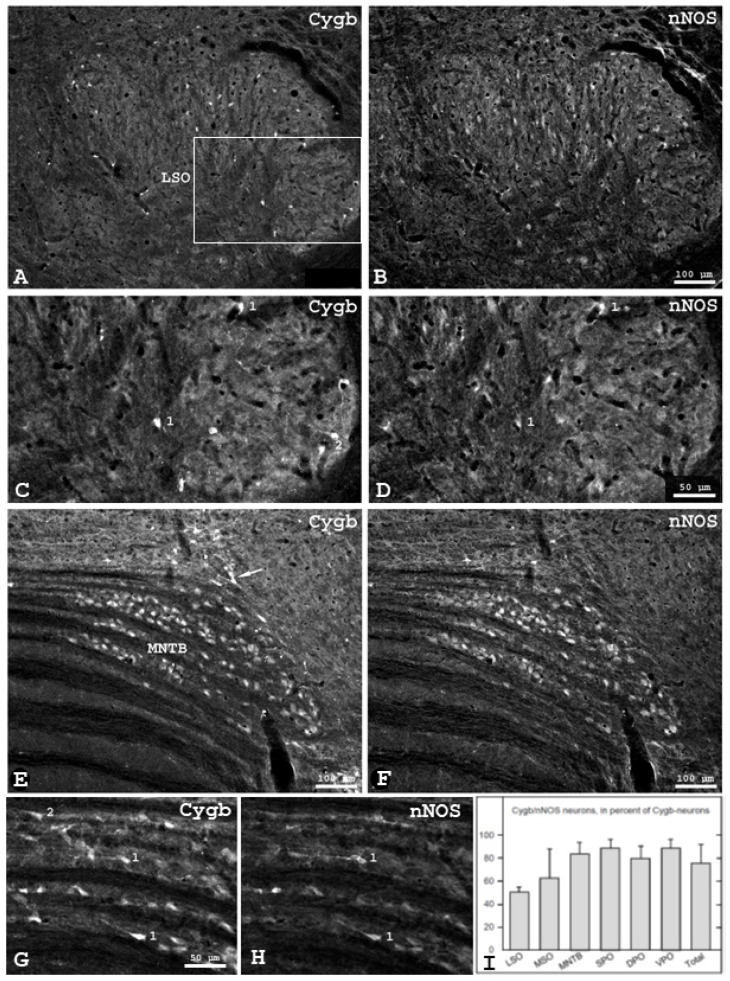
Distribution of cytoglobin (Cygb) and neuronal nitric oxide synthase (nNOS) immunofluorescence in the right SOC of the mouse (frontal plane). (**A**,**B**): Some immunoreactive neurons are seen in LSO and in periolivary regions. The boxed region in (**A**) is shown in higher magnification in (**C**,**D**). Double-immunoreactive neurons in the lateral superior LSO region (exemplarily labelled with "1" in (**C**,**D**)), and neurons expressing solely Cygb (“2”) or nNOS were present. (**E**,**F**): Cygb- and nNOS-immunofluorescence in the right medial nucleus of the trapezoid body (MNTB) of the mouse. The majority of large neurons show colocation of Cygb and nNOS. Strikingly, a group located at the dorsal edge of the MNTB (arrow in (**E**)) contains predominantly Cygb-positive, nNOS-negative neurons. (**G**,**H**): Magnifications of the medial portion of the MNTB in an adjacent section show both substances colocated in some neurons (exemplarily marked with "1"), while others ("2") contain only Cygb. (**I**): Double-labeled neurons in the SOC subregions in percent of Cygb-neurons (mean ± SD, *n* = 5).

## Data Availability

The data that support the findings of this study are available from the corresponding author, Stefan Reuss, upon request.

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
