# Peer review of "Neuronal Cytoglobin in the Auditory Brainstem of Rat and Mouse: Distribution, Cochlear Projection, and Nitric Oxide Production"

_brainsci, 2023, doi:10.3390/brainsci13010107_

Round 1

Reviewer 1 Report

1. The article type should be mentioned in the title.

2. Abstract – the abstract should be improved

a. Remove references and citations.

b. Provide the objective of the study.

c. Provide methodology

d. Provide results encountered in numbers (percentages, …)

3. Methods

a. IRB (Ethical committee) number should be provided

4. Statistical analysis

a. a section about statistical analysis should be provided

5. Results

a. The inclusion of references in the results should be avoided. Please, restructure the paragraphs or request permission from the editor.

6. Discussion

a. a section about the limitations of the study should be done.

7. Conclusion

a. relocate the abbreviations.

Could the authors provide a figure (schematic diagram) explaining the proposed mechanisms? The reviewer believes that this would significantly impact the quality of the manuscript.

Reviewer 2 Report

This is a well-structured research article. The main question addressed by this research is cytoglobin expression in the superior olivary complex of the auditory brainstem of rat and mouse. I think that this is an original and interesting topic for the readers of this journal.

This paper adds to this scientific area as it is the first analysis of cytoglobin expression in the superior olivary complex of the auditory brainstem of two rodent species.

The introduction gives the background of this study as it describes the cytoglobin protein, its role and the lack of adequate relative literature data.

“Materials and Methods” section is descriptive enough and well-structured too. It refers to the animals used, ethical aspects, and the processes that were followed in order to have the localization of cytoglobin protein in the brains of rat and mouse depicted.

The results are very interesting and, to my opinion, well presented.

The discussion is well written, summarizing and discussing the main findings of the study; perhaps a paragraph summarizing the limitations of this study would further improve the scientific value of this paper.

Conclusions although consistent with the evidence presented, perhaps could be written in a more detailed manner, presenting and analyzing some specific targets for future studies, that have been mentioned in the discussion.

References are relative to the subject and sufficient in number.

English language and style are generally fine.

Reviewer 3 Report

The present study is a continuation of a previous one reported by the authors, in which the cytoglobin expression was reported in auditory brain areas of the mouse such as the CN, the SOC, the IC and the MGB. Here, authors focused on the Cygb distribution in the SOC of the rat and mouse. They reported that approximately 10% of the SOC neurons were immunopositive for Cygb, half of those neurons were olivocochlear neurons. To do so, the authors combined neuronal tract-tracing experiments (injecting a pure retrograde tracer-FG- into the cochlea) with immunofluorescence to detect Cygb and neuronal nitric oxide synthase (nNOS). The manuscript is well written and easy to follow, being scientifically sound and providing relevant information. However, I have several concerns regarding the methodology and I pointed out the weakest points of the study with the hope that the authors can improve the quality of the article. Also, I found some minor errors that are easy to correct.

One of the weakest points of the article is that the visualization of the results is very poor. The authors did not used different colors to differentiate between the Cy3 and Cy2 fluorochromes or the Fluorogold. Without this, they cannot generate merge images and it is very difficult to visualize the double labeling neurons. Therefore, the quantification analyses are not very reliable, since the authors cannot clearly show the colabeling. In other words, the authors cannot be sure they are counting all the immunolabeled neurons (particularly those showing Cygb and nNOS colocalization) by using an old-fashion way to visualize the fluorescence. An epifluorescence microscope is not the proper equipment to use for co-localization analysis. The authors should use a scanning confocal microscope at least to show the distribution of Cygb and nNOS in the neurons, obtaining merge images with the fluorochromes in different colors and an orthogonal view of the sections to detect the internal distributions of the proteins. I strongly recommend using confocal microscope images to visualize and carry out the quantification analyses of double-labeling neurons, and hence the results will be more convincing and the quality of the study will be notably improved.

The authors quantified single- and double-labeled neurons separately and the respective percentages were calculated for each SOC nucleus. It is not very clear the criteria used to count cells. Do the authors make sure that they are not counting the same cell in different sections? How can the authors be sure of miscounting immunoreactive cells that were nearby? It seems that the magnification used in the images shown in the manuscript might not be enough to clearly differentiate between nearby cells. What was the lens objectives (magnification) used in the images for the quantification analyses? This important information is missing and implies that it was not clear how the authors used the Abercrombie correction in the quantification analysis. According to the authors Cygb can labeled both neuronal somata and the cell nucleus. However, this immunolabeling feature is not clearly shown in the manuscript since the authors did show colocalization of Cygb with a proper marker for nuclear staining (e.g. DAPI). Therefore, the cellular distribution of Cygb must be study further and clearly showed in high magnification images. In addition, I have several concerns on how the quantification was made. Did the quantification was performed by only one researcher? If it was the case, the researcher was always the same? how many images (sections) of each animal were used for quantification? Did the authors include for the quantification all the rostro-caudal sections of the SOC?  If this was the case, did the authors found differences in the rostral-caudal distribution of Cygb and nNOS-immunolabeled neurons? It strongly recommended to show a rostro-caudal scheme of the rat and mouse SOC nuclei with the corresponding bregma coordinates, showing the distributions of double-immunolabeled neurons as well as those labeled with FG. The authors can use this sketch to give an overview/summary of the results and compare between the labeling shown between the cases. A sketch for the SoC mouse and another for the rat can be very useful to compare between the two species.

Lines 177-178 “When selected sections were exposed to fluorescent Nissl counterstain, the morphological appearance of Cygb-neurons, such as soma size and number and shape of processes, did not differ from neighboring Cygb-negative neurons”. The author should show these results with an image. There is not mention in the methodology regarding the fluorescent Nissl staining. How do they do it? Did the authors carry out the Nissl staining in alternating sections? What is the catalog number of the products used in the Nissl protocol? This information is missing in the methodology and it is not shown in any of the figures.

The authors stated in the manuscript that the Cygb-immunoreactivity was very restricted in neuronal regions. However, the low magnification image of a brainstem coronal section shown in Fig. 1A indicated that the Cygb-immunolabeling is in many brain regions. I can see many Cygb-immunolabeled neurons in the reticular formation and nearby the pyramidal tracts. Indeed, the Cygb-immunolabeling seems stronger in the brainstem reticular formation that in the SOC nuclei. It seems that the specificity of Cygb-immunoreactivity is not the one claimed by the authors. Can the authors clarify this?

Minor errors…

Line 74…the bracket is missing after the references.

Point 2.1. The approval numbers of the research projects/protocols provided by the institutional bioethical committee is missing in the article as well as the referenced european laws related to the use of laboratory animals.

Although the surgical approach was identical to that used in previous studies, the reader needs a brief summary with some information about the process.

Line 95…What the authors mean with “at the middle of the light period”? when the animals were euthanized?

Line 102…”frontal plane” is not anatomically correct. Change “frontal plane” to “coronal plane”.

Line 104….the words corresponding to the abbreviation “BSA” are not in the text of the manuscript.  

Line 106…The authors should indicate that the N-terminus is that of the “Cygb protein”.

Figure 1….The authors should depict in the low magnification image (Fig 1. A) the squares corresponding to the high magnification images (Figs. 1 C-G), similarly to Fig. 2 and Fig. 3.  

Round 2

Reviewer 1 Report

I would recommend the following changes in the manuscript that I did not observe.

1. The article type should be mentioned in the title.
2. Abstract – the abstract should be improved---Remove references and citations.
4. Statistical analysis---a section about statistical analysis should be provided
5. Results---The inclusion of references in the results should be avoided. Please, restructure the paragraphs or request permission from the editor."

Reviewer 3 Report

The current version of the manuscript contains more information and the modifications relating to contents and methods add clarity and understanding. The authors addressed most of my concerns, however they totally discarded the use of LSM and I do not agree with the reasons provided for the authors for doing so. I can understand that the authors are unwilling to use the LSM for many arguments, but they cannot argue that the LSM is significantly better to visualize and quantify double-labeled neurons than an epifluorescence microscope. In the present study, the authors indicated that the LSM method is not well suited for the quantification of larger regions. I am quite surprised by this response. Indeed, LSM are equipped with software that created mosaic images that allows to pinpoint and count accurately the colabeled neurons in whole brain sections. Therefore, I disagree with the statement “it is impossible” as the author said. I know very well the epifluorescence microscope used by the authors, and I am not saying that it is not valid to answer the question raised by the authors in this paper. I said that if the authors want to demonstrate the colocalization between Cygb/FG or Cygb/nNOS, they must show it in an orthogonal view (projection) from z-stacks of a confocal microscopy as it is commonly done. Otherwise, the results no longer convince anyone. The renders of the widefield epifluorescence microscope are unsuitable for colocalization analysis (Dunn, K. W., Kamocka, M. M. & McDonald, J. H. A practical guide to evaluating colocalization in biological microscopy. Am. J. Physiol. Cell Physiol. 300, C723–C742, 2011). Therefore, I strongly recommend showing at least one z-stacks image to verify that the florescent labels are located in the same neuron or near to one another.

Figure 3E-F serve as an example of my concern. It is not clear at this very low magnification that the cell depicted with an arrow in Fig3E-F is a nNOS-negative neuron as claimed by the authors.  It seems there is also immunolabeling for nNOS in this cell (see Fig 3F). Indeed, it seems there are more than one cell. In other words, the image is very poor to clearly show the findings claimed by the authors. There will be no doubt using z-stack confocal image at higher magnifications (40x or 63x objective lenses).

Why are the authors not showing confocal merge images to visualize colabeling neurons? Is this because of readers with deuteranopia? The authors can create a pseudocolored confocal image very easily to make the visualization friendly for these readers. Indeed, it is very frequent to do so in magenta color for publication as it is a requirement in many journals. Therefore, this seems a convenient excuse rather than an impediment for using the confocal microscope.

The authors said, “The objective magnification (usually x20) is only one aspect of image generation. There are tubus and other factors, and modern microscopes with digital color cameras are connected to displays from which analysis (and discussion with colleagues) is conducted”. In my opinion, the parameters, and settings (including the objective magnification) that were used to take the microscope images for the qualification analyses or visualization must be obligatory in the material sections of the manuscript. This information is essential for reproducibility of the results.

The authors said, “There were basically no differences apparent in the rostro-caudal extend and there were basically no differences between rat and mouse, apart from the lower number of neurons, which fits well with differences in brain size”. This data might be interesting in SOC nuclei that contain heterogeneous populations of neurons like the VNTB or LSO that have tonotopic distributions within the nucleus. Maybe, it is worthy to mention this observation, but I understand that the authors should decide the convenience to do so.  

Remove “5” in line 457.

The manuscript has been improved, and my intention is to help the authors to improve the visualization of the results and the quality of the paper, and I hope that they take my comments in a constructive manner.
